



# Technical note: Darkroom lighting for luminescence dating
# laboratory
Marine Frouin[1], Taylor Grandfield[1], William Huebsch[1], Owen Evans[1]
[1]Department of Geosciences, University of Stony Brook, USA
*Correspondence to*: Marine Frouin (marine.frouin@stonybrook.edu)
**Abstract.** An optimal lighting setting for the darkroom laboratory is fundamental for the accuracy of luminescence dating
results. Here, we present the lighting setting implemented in the new Luminescence Dating Research Laboratory at Stony
Brook University, USA. In this study, we performed spectral measurements on different light sources and filters. Then, we
measured the optically stimulated luminescence (OSL) signal of quartz and the infrared stimulated luminescence (IRSL) at
50°C ($IR_{50}$) as well as post-IR IRSL at 290°C ($pIRIR_{290}$) signal of potassium (K)-rich feldspar samples exposed to various
light sources and durations.
Our ambient lighting is provided by ceiling fixtures, each equipped with a single orange light-emitted diode (LED). In addition,
our task-oriented lighting, mounted below each wall-mounted cabinet and inside the fume hoods, is equipped with a dimmable
orange LED stripline.
The ambient lighting, delivering 0.4 lux at the sample position, induced a loss of less than 5% (on average) in the quartz OSL
dose after 24 h of exposure, and up to 5% (on average) in the $IR_{50}$ dose for the K-rich feldspar samples, with no measurable
effect on their $pIRIR_{290}$ dose. The fume hood lighting, delivering 1.1 lux at the sample position, induced a dose loss of less
than 5% in quartz OSL and K-rich feldspar $IR_{50}$ doses after 24 h of exposure, with no measurable effect on their $pIRIR_{290}$ dose.
As light exposure during sample preparation is usually less than 24h, we conclude that our lighting setting is suitable for
luminescence dating darkrooms, it is simple, inexpensive to build, and durable.
**1 Introduction**
Luminescence dating techniques enable evaluation of the time that has elapsed since crystallized mineral grains, such as quartz
and feldspar, were last exposed to sunlight or high temperature. Hence, a fundamental requirement of the method is that the
light-sensitive traps in mineral grains must have been entirely emptied in the past and remained unexposed to light until
laboratory measurement (Aitken, 1998). During sample collection in the field and sample preparation in the laboratory,
precautions should be taken to preserve the integrity of the samples using controlled lighting conditions; otherwise, there is a
severe risk of reducing the dating signal (i.e., luminescence signal) and hence the apparent age (i.e., deposition time) of the
mineral grains. For quartz grains, the shorter wavelengths (less than 360 nm) are most effective in evicting electrons from traps
(Aitken 1998, Fig 8.11). For K-rich feldspar grains, the bleaching resonance is centered at 860 nm. For quartz and feldspar
grains, dim lighting conditions in the orange-yellow to red wavelength provide minimal signal loss over a limited time (Aitken,



1998). Within this large wavelength range, each luminescence dating laboratory worldwide defines its lighting conditions. In
fact, only a few laboratories have reported measurements of their lighting conditions (e.g., Spooner, 2000; Huntley and Baril,
2002, Lindvall et al., 2017; Sohbati et al.,2017, 2021) and their effect on the mineral samples.
Here we report on the lighting conditions implemented in the new Luminescence Dating Research Laboratory at Stony Brook
University. First, we performed spectral measurements on different light sources and filters. Then, we measured the dose loss
of quartz and potassium (K)-rich feldspar samples after exposure to various light sources and times.

## 2 Instrumentation

Spectral measurements were performed using a Qmini Wide VIS (AFBR-S20M2WV) spectrometer with a spectral range of
212–1035 nm (sensitivity optimized at ~500 nm) and a spectral resolution at 1.5 nm equipped with an optic fiber P400-1-UV-
VIS400. The calibration of the spectrometer was performed in May 2019. All spectra were measured over a total integration
time of 2 s. The amount of light on the laboratory benchtops was measured with a luxmeter Dr.meter LX1330B digital
illumination/light meter.

43         Calibration quartz (180-250μm, batch #118 and #123 from Risø), as well as quartz (SB27) and feldspar grains

(SB36 and SB44) extracted from natural samples, were used in this study. The natural samples were selected from the Stony
Brook Luminescence Dating Research Laboratory collection. Sample SB27 was collected from the middle palaeolithic site
of Oscurusciuto (Italy). Samples SB36 and 44 were from the last glacial cycle and collected on Long-Island, NY.
Coarse grain (180-250μm) fractions were dispensed on 10-mm-diameter aluminium discs (quartz) and cups (feldspar) with a
silicone oil adhesive of 4 mm diameter. Sixty aliquots per sample were prepared.
The luminescence measurements were performed on a Risø TL/OSL DA-20 reader equipped with a photomultiplier tube ET
PDM9107-CP-TTL and a $^{90}Sr/^{90}Y$ source delivering a dose of $0.106 \pm 0.003$ Gy.s$^{-1}$ to the material deposited on a disc. The
luminescence signal from the quartz grains was stimulated with blue diodes emitting at 470±30 nm and detected through a
combination of a 2.5- and 5-mm-thick Hoya U-340 glass filters (transmission between ~290–370 nm). The infrared stimulated
signal from the K-rich feldspar grains was stimulated with LEDs emitting at 850±30 nm, and the luminescence signal was
detected through the so-called blue filter pack composed of a 3mm thick Schott BG3 and a 2-mm-thick Schott BG39 filter
(detection window centred on 410 nm).
A standard multi-grain Single-Aliquot Regenerative (SAR) procedure was used for the dose determination. After the
measurement of the natural OSL signal, the aliquots were subjected to regenerative-dose cycles (including a duplicate dose
and zero dose). The quartz OSL signal was measured for 40 s at 125°C prior to heating at a higher temperature for the quartz
samples. The net intensity of the blue luminescence signal was integrated over the first 0.8 s after subtracting the background
signal derived from the last 8 seconds of stimulation. For feldspar, equivalent doses were measured using SAR protocols
exploiting the IRSL signal measured at low temperature and referred to as the IR$_{50}$ protocol (Huntley and Lamothe, 2001), as





well as the post-infrared-infrared luminescence signal measured at high-temperature, and referred to as the pIRIR$_{290}$ (Thiel et
al., 2011). Both luminescence signals were integrated over the first five seconds of stimulation, and the background was taken
from the last 10 s of stimulation. For quartz and feldspar samples, the growth curve was fitted with a single saturating
exponential function. The uncertainties on an individual dose have been determined using classical rules of error combination
using the Analyst software (Duller, 2007), a further systematic uncertainty of 2% was added in quadrature to each uncertainty
value to account for calibration errors and machine reproducibility.
**3. Methodology**
**3.1 Lighting condition**
The decay of luminescence in both quartz and feldspar can be induced by any wavelength of solar radiation. More precisely,
the maximum bleaching rate of the quartz OSL signal is induced by short wavelength (in the UV-blue-green region), while
feldspar IRSL signals have their bleaching resonance in the long wavelengths (in the red-infrared region). Therefore, finding
an optimum lighting condition for both quartz and feldspar is difficult. Some luminescence laboratories use red bulbs or red
fluorescent tubes, which are particularly well adapted for quartz (Sutton and Zimmerman, 1978). Lamothe (1995) reports that
restriction to the wavelength region 650-600 nm can be obtained from a white fluorescent tube using three layers of Lee 106
filters (i.e., deep red) and an infrared trimming glass filter. However, Lindvall et al. (2017) reports a loss of 3 to 21% of the
quartz luminescence signal intensity after 24 h of exposure to the red wavelength. For feldspar, there is an optimum at 620-
540 nm in the yellow part of the spectrum (Huntley and Baril, 2002, their Fig.1). Orange-yellow wavelength can be obtained
using a low-pressure sodium vapor lamp with appropriate yellow filters to block the blue to ultraviolet emissions (Spooner,
1993, 2000). Sohbati et al. (2017, 2021) also observed that using amber light-emitting diodes (LEDs) with an emission peak
at 594 nm, quartz and feldspar lost only between 1 to 3 % of luminescence signal intensity after 48-hour of exposure.
On another note, a comfortable laboratory illumination level is required for the safety of those spending long hours working
in the darkrooms. In low light conditions (e.g., moonless night), human eyes have a maximum sensitivity at 507 nm (in the
blue-green region), and red light is almost invisible. Green wavelength cannot be used in our laboratory as our lighting
environment, as it bleaches the quartz OSL signal. However, the closest solution and, therefore our best compromise is the
orange-yellow wavelength, similar to what was recommended by Sobhati et al. (2017, 2021).
**3.2 Bleaching test procedure**
All aliquots were bleached for five days in a solar simulator (UVACUBE400) equipped with a SOL500 lamp filtered with an
H1 filter glass (transmission range from 315 nm to 800 nm). Quartz samples received an artificial beta dose of 5Gy (calibration
quartz), 20 Gy (natural quartz samples), or 69.7 Gy (K-rich feldspar samples). A recovery dose test was performed on a series
of three aliquots for each sample. All the aliquots were placed at different locations in the darkrooms for 24, 72, 240, and 720



hours, and their remaining dose was measured and normalized by the recovery dose. Noting that 720h exposure is an unrealistic
exposure time for sample preparation in the laboratory, nevertheless, we wanted to investigate the effect of extremely long
exposure.
To monitor the bleaching effect of the ceiling fixtures, the aliquots were placed on a benchtop at a workstation. To monitor the
bleaching effect of the dimmable LEDs, we fixed the light intensity at 20 % and 30 % of their maximum intensity inside our
two fume hoods with a black benchtop, and at 20% inside our fume hood with a white benchtop.
In nature, the quartz OSL signal bleaches faster than the K-feldspar signals, and the K-feldspar $IR_{50}$ signal bleaches faster than
the K-feldspar $pIRIR_{290}$ signal. Therefore, the OSL and $IR_{50}$ signals are key for monitoring the bleaching effect of our
laboratory darkroom lights, rather than the $pIRIR_{290}$. The $IR_{50}$ signal is, however, and contrary to the $pIRIR_{290}$ signal, affected
by anomalous fading, which is a loss of luminescence signal through time. To account for fading and overcome any laborious
fading correction, we measured all the aliquots 720 hours after the initial beta irradiation. In practice, a set of aliquots was
given a dose of 69.7 Gy, and then stored in the dark for 720 hours, while another set of aliquots was exposed to a light source
for 24 hours and then stored in the dark for 696 hours, while another set of aliquots was exposed for 72 hours, and then stored
in the dark for 648 hours, and so on. Assuming that all the aliquots are affected by the same fading rate after one month, any
tendency that we will observe as a result of our bleaching test is assumed to be the only effect of the light exposure.
**4 Results**
**4.1 Spectral analysis**
We measured the emission spectrum of three light sources: a red LED PAR38, a deep orange single LED, and a
dimmable deep orange LED stripline. Details on the LEDs are reported in Table 1. The PAR38 LED emits a peak wavelength
at ~600 nm (FWHM ~84 nm) with a large tail in both the short and the long wavelength emissions and a low-intensity peak at
~452 nm, in the blue region of the spectrum (Fig. 1a). The single LED emits a peak wavelength of 594 nm and the stripline of
LEDs emits a peak wavelength at 596 nm (Fig. 1a). Both peaks are narrow with a FWHM of ~ 14 nm. Contrary to the red PAR
38 LED, the single and stripline LEDs results are the closest to our preferred conditions.

**Table 1.** LED details given by the manufacturers.

| Type | Name | Lumens | Wavelength (peak) | Wavelength (dominant) | FWHM | Viewing angle | CIE xy | Company (ref) |
|------|------|--------|-------------------|-----------------------|------|---------------|--------|---------------|
| Ambient | Cree XLamp XP-E2 LEDs | Flux: 73.9 lm (min.) @ 350mA | 590 nm | 590 nm | 5 nm | 110 | - | LEDsupply (CREEXPE2-COL-X 1-Up) |

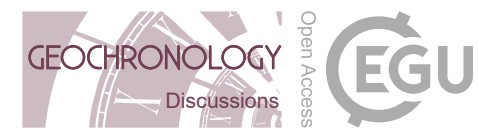

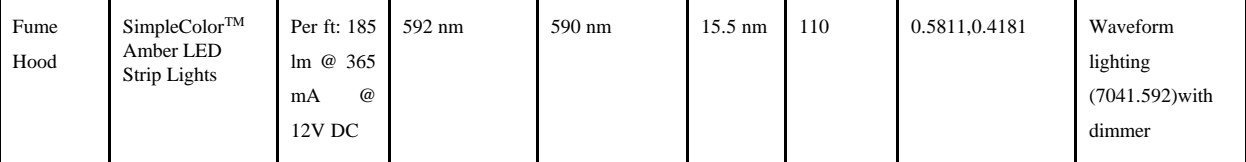

| Fume Hood | SimpleColor™ Amber LED Strip Lights | Per ft: 185 lm @ 365 mA @ 12V DC | 592 nm | 590 nm | 15.5 nm | 110 | 0.5811,0.4181 | Waveform lighting (7041.592)with dimmer |
|---|---|---|---|---|---|---|---|---|

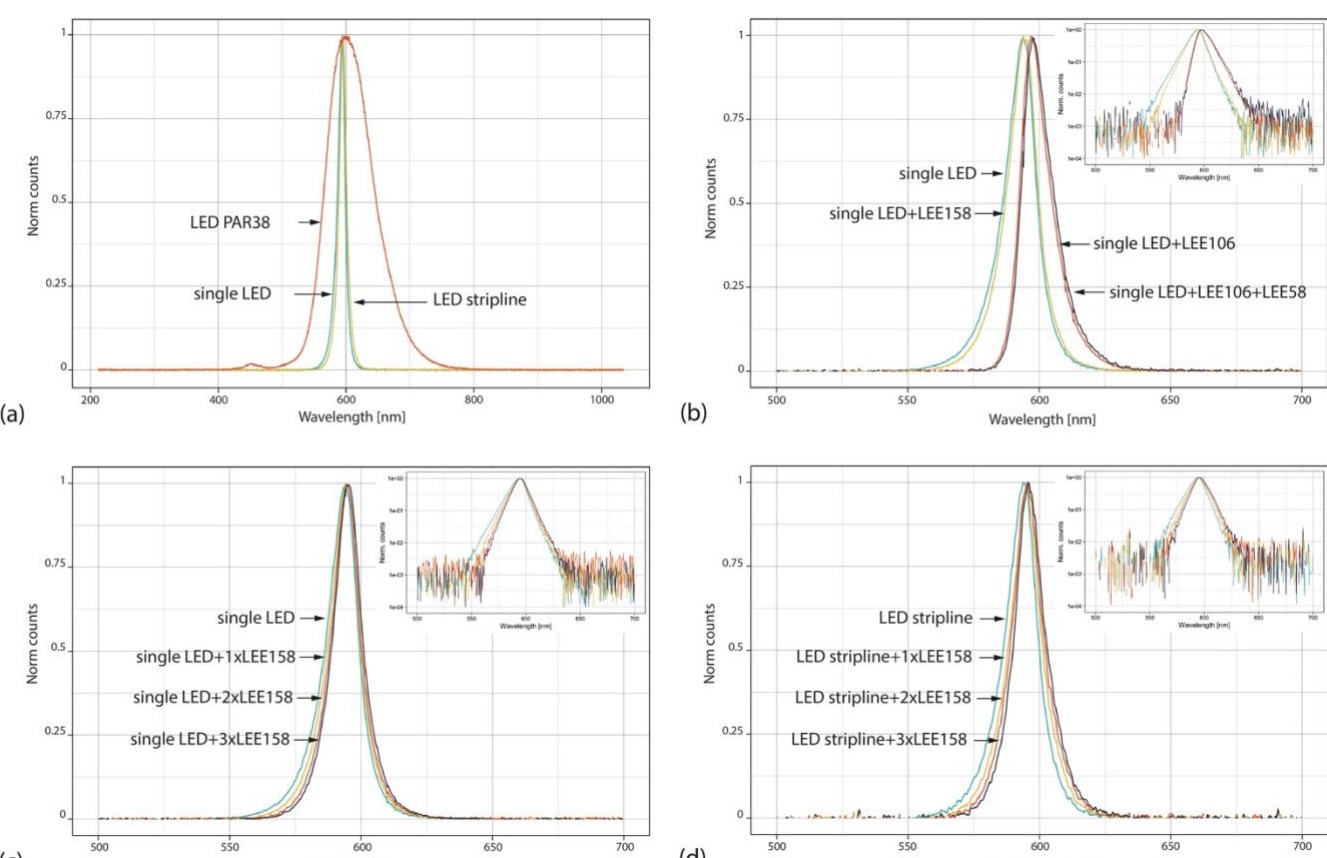

(a)      (b)      (c)      (d)

**Figure 1: The normalized emission spectra of (a) the LED PAR38, single LED, and LED stripline, (a) the single LED though different long-pass filter combination, (c) the single LED through layers of the 158 Deep Orange LEE filter (LEE158), (d) the LED stripline through layers of the LEE158.**

The single LED and the stripline LEDs have, however, a tail in the short wavelengths starting at ~530 nm in the green region of the spectrum. To reduce this short wavelength emission, we measured the emission spectrum of the single LED with a series of long-pass filters: 106 primary red LEE, which has a cut-off at 580 nm, and 158 Deep Orange LEE, which has a cut-off at 530 nm. As expected, the primary red filter successfully removed the short-wavelength emission (Fig. 1b), however, the peak wavelength shifted from 594 nm to 597 nm, and a tail in the long wavelength emissions appeared (up to 640 nm). With



the orange filter, the tail in the short wavelengths is slightly reduced, while the rest of the LED emission spectrum remains the
same (Fig. 1b). Using both filters simultaneously results in an emission spectrum similar to the one obtained with the primary
red filter (Fig. 1b). In order to narrow the emission band of the single LED, we measured its spectrum with additional layers
of 158 Deep Orange LEE long-pass filter. Figure 1c shows that adding one, two, or three layers of orange filter significantly
contributes to reducing the short-wavelength emission while slightly increasing the long-wavelength emission. With three
layers of orange filter, the single LED peak wavelength is at 595 nm (FWHM ~13 nm). Similarly, adding three layers of 158
Deep Orange LEE long-pass filter in front of the stripline LEDs successfully removes the green emission (Fig. 1d), while the
peak emission remains at 596 nm (FWHM ~13 nm).
137          We decided to use single LEDs in our ceiling light fixtures. The ceiling lighting consists of line track fixtures made
of aluminium alloy placed at ~2.6 m from the floor (Fig. 2a). Each fixture has a single orange LED covered by three layers
of 158 Deep Orange LEE filters and a transparent acrylic glass (1mm thick). We checked that the transparent acrylic glass
does not change the light spectrum. Inside the fume hoods, we used the dimmable LED stripline covered by three layers of
158 Deep Orange LEE filters and a transparent acrylic glass (3 mm thick) (Fig. 2b-d). The same stripline of dimmable
orange LEDs with 158 Deep Orange LEE filters was fixed under the wall-mounted cabinets.





**Figure 2: Pictures of the laboratory setting in the laboratory darkroom showing the ceiling light fixture (a), and the fume hood lighting (b-d).**

**4.2 Bleaching test**



147        Here we report on the capacity of our light sources in bleaching quartz and feldspar samples. Each ambient fixture

delivers 0.4 lux at the sample location on a benchtop. The intensity of the LED stripline in fume hood #1 was fixed at 20%
and delivered 1.1 lux at the sample location on a white benchtop (referred to as I=20% WB in Fig. 3). The intensity of the
LED stripline in fume hood #2 was fixed at 20% and delivered 1.1 lux at the sample location on a black benchtop (referred
to as I=20% BB in Fig. 3). The intensity of the LED stripline in fume hood #3 was fixed at 30% and delivered 1.7 lux at the
sample location on a black benchtop (referred to as I=30% BB in Fig. 3). These settings remained constant throughout the
experiment.

**Figure 3: Ratio between the measured OSL dose from aliquots exposed to light and the measured dose from aliquots unexposed. The figures show the results from (a) the Risø calibration quartz exposed to the ceiling light fixture, (b) the quartz sample SB27 exposed to the ceiling light fixture, (c ) the Risø calibration quartz exposed to fume hood lighting, and (d) the quartz sample SB27 exposed to fume hood lighting. Three aliquots were measured per exposure time. The long dashed line indicates a ratio of 1, and the dashed line indicates a loss of 5%.**





160         Figure 3a-b shows the dose decrease after exposure to the ceiling light fixture for the Risø calibration quartz and

sample SB27. Both samples displayed a ~3% (average) dose loss after 24 h and ~5% after 72 h. After a substantially longer
exposure of 720 h, the Risø calibration quartz displayed a dose loss of ~10% and sample SB 27 of ~18%. Figure 3c-d shows
the remaining dose after exposure to the LED striplines within the fume hoods. For the Risø calibration quartz, the signal
loss is indistinguishable for the three settings after 24 h exposure (less than 5%). Beyond this time, however, the fume hood
with the LED set to an intensity of 30% induced the fastest signal loss. The bleaching rates between the fume hood with the
light intensity fixed at 20% and the white benchtop or the black benchtop are indistinguishable. For both settings, the dose
lost is ~1% after 24 h exposure and ~10% after 720 h exposure. For quartz sample SB27, a similar tendency has been
observed; a signal loss of ~1 % (average) has been recorded for the three settings after 24 h exposure. For the fume hoods
with the light intensity fixed at 20%, a ~10% loss in dose was recorded after 240 h exposure, and up to 18% after 720 h. The
light fixed at 30% intensity provoked the fastest dose loss.
This set of measurements has been repeated on two K-rich feldspar samples. The results show more dispersion in the measured
dose, probably due to the anomalous fading (all aliquots were stored and/or exposed for 30 days before measurement). Figure
4a-b illustrates the remaining dose after exposure to the ceiling light fixture. After 72h of exposure, the dose loss is up to ~5%
for both samples. After this, there was a dramatic drop in signal for both samples. After 720 h exposure, the dose loss is
between 30 to 40%.



**Figure 4: Ratio between the measured $IR_{50}$ dose from aliquots exposed to light and the measured dose from aliquots unexposed.**
**The figures show the results from (a) the feldspar sample SB36 exposed to the ceiling light fixture, (b) the feldspar sample SB44**
**exposed to the ceiling light fixture, (c) the feldspar sample SB36 exposed to fume hood lighting, and (d) the feldspar sample SB44**
**exposed to fume hood lighting. Three aliquots were measured per exposure time. The long dashed line indicates a ratio of 1, and**
**the dashed line indicates a loss of 5%.**



Figure 4c-d shows the remaining dose of the initial given dose after exposure to the LED striplines within the fume
hoods. The LED's set to an intensity of 30% displayed the most rapid dose loss. After 24 h of exposure, both samples lost
between 5 to 10% dose, and up to ~40 to 60% after 720 h exposure. For the settings set at 20% intensity, there was no loss
dose recorded for sample SB36, after 24 h of exposure. The dose loss remains less than 5 % after 72 h of exposure and less
than 10% after 240 h. After 720 h of exposure, the dose loss ranges between 20 to 40%. For sample SB 44 (Fig 4d), the aliquots
exposed to the LED stripline with an intensity of 30% had a ~10% dose loss after 24h, and ~60% dose loss after 720 h of
exposure. For the aliquots placed under the fume hoods with an LED intensity of 20%, the dose loss was up to 5% after 24 h,
10% after 72 h, and between 30 to 40% after 720h.
The experiments have been repeated to measure the bleaching effect of each setting on the $pIRIR_{290}$ dose of the same
K-feldspar samples (SB 36 and SB 44) for up to 72 h of exposure. The measured De's are undistinguishable with the given
dose at 1 sigma, and therefore indicate no measurable bleaching effects of our light sources on the $pIRIR_{290}$ signal.
**5 Conclusion**
Two lighting settings have been implemented in the new Luminescence Dating Research Laboratory at Stony Brook
University. For ambient lighting, ceiling fixtures were equipped with single orange LEDs. For task-oriented lighting, a
dimmable orange LED stripline was mounted below the wall-mounted cabinets and inside the fumehoods. Both settings are
covered with three layers of 158 Deep Orange LEE filters, and their peak wavelength is at 595 nm and 596 nm, respectively.
Our bleaching tests quantified the dose loss in quartz and K-rich feldspar samples with exposure. The ambient lighting
delivering 0.4 lux at the sample position induced a loss of less than 3% in the quartz OSL dose after 24 h of exposure, and up
to 5% in the $IR_{50}$ dose for the K-rich feldspar sample, with no effect on its $pIRIR_{290}$ dose. The fume hood lighting at an intensity
of 20%, delivering 1.1 lux at the sample position, induced a loss of less than 5% in quartz OSL and K-rich feldspar $IR_{50}$ doses
after 24 h of exposure. At a higher intensity of 30 %, the stripline of LEDs induced more rapid bleaching. Then, we recommend
using it only in case of emergency or during lab cleaning.
Our setting is well adapted to luminescence dating darkrooms by providing a comfortable laboratory illumination for the
operator, which has a minimal bleaching effect on the samples. During laboratory preparation, the samples are exposed to
ambient lighting only for a few hours, mainly during sieving and density separation, and to the fumehood lighting for a few
minutes when pouring chemicals. The total light exposure to darkroom lighting should be less than 24h. In addition, extreme
precautions should be taken at each step to avoid unnecessary light exposure by using non-transparent beakers when possible,
covering the sample container with an opaque lid or aluminium fold, switching off the light in the fume hood when sample
manipulation is not necessary, and storing the sample as long as possible in an opaque container while preparing the aliquots.
Finally, we plan on monitoring regularly the bleaching effect of our light sources as we work on samples from various origins.




**Code/Data availability:** All data are available upon request.

**Author contribution:** MF designed the experiments, and TG carried them out. WH and OE built the light ceiling fixture. MF designed and built the LED striplines.

**Competing interests:** We declare no competing interests

**Acknowledgments:** MF would like to thank Desmond DeLanty (architect) for designing the ceiling fixtures and installing all the light sources in the laboratory.

219

220

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

measured the loss of signal in quartz and feldspar samples when exposed to various light sources and durations. Finally, our
lighting setting is suitable for a luminescence darkroom laboratory, it is simple, inexpensive to build, and durable.