# Peer review of "Technical note: Darkroom lighting for luminescence dating laboratory"

_Geochronology, 2023_

## Author Response (AR1)

Dear Sumiko,

Thank you for this thorough review. We have considered all your comments in the revised version of our manuscript and took into consideration all the reviewer's suggestions.

Line 28 onwards: you refer Aitken (1998) about bleaching of quartz and feldspar. But there is no Fig. 8.11 in the book. The closest diagram I found is Fig. 6.6 for quartz and Fig. 6.7 for feldspar. But there is no data below 400 nm in Fig. 6.6. Please check the information source and modify the sentences.
>> Well spotted. This is a mistake.

Line 43: here there is no space between "180-250" and micron, whereas most of other places there is a space between a number and a unit. Please be consistent throughout the text.
>> This has been corrected

Line 54: similar to the point above, "3 mm think" or "3-mm-thick"? Please check throughout the text.
>> Done

Line 56-67: Please add preheat conditions for the three measurement protocols for quartz and feldspar.
>>l.62 we added the following sentence:" The SAR protocol was applied to quartz samples with a preheat of 220°C for 10 s, and a cutheat of 180°C".
>> l.79 we added: "Prior to the IRSL stimulation, standard preheat conditions were applied at 250°C for 60 s and 320°C for 60 s, for the $IR_{50}$ and $pIR-IR_{290}$ protocols, respectively."

Line 90: "natural quartz samples" and "K-rich feldspar samples": I would use sample codes instead. Especially "natural quartz" would be misleading, because it could give an impression that 20 Gy is the natural dose. Please round 69.7 Gy to 70 Gy.
>> Agreed. This paragraph has been rewritten and now reads as follows:"In this study we used two quartz samples and two feldspar samples. One of the quartz sample is the calibration quartz (180-250 µm, batch #118 and #123; Hansen et al., 2015). The second quartz sample (SB27) was collected from the middle palaeolithic site of Oscurusciuto (Italy) and had a natural average dose of 133±5 Gy (n=14) (publication in prep). The feldspar samples SB36 and 44 were from the last glacial cycle and collected on Long-Island, NY. Sample SB36 had a saturated $pIR-IR_{290}$ dose ($2D_0$=328±10 Gy, n=3). Sample SB44 had an average $pIR-IR_{225}$ dose of 49±1 Gy (n=11, not fading corrected) and a $pIR-IR_{290}$ dose of 67±3 Gy (n=12)."

Line 90: "A recovery dose test" is also misleading. It is not a dose recovery test.
>> Agreed. This has been corrected.

Line 160-175: both signal intensity loss and dose loss are mentioned here, but the data shown in the paper are only in the measured dose ratio. Is the loss of signal intensity almost equal to the loss in dose or not?
>>The signal loss is equal to or slightly lower (up to 2%) than the dose loss.

L.237, we added "For all samples, we decided to report the results as dose loss because such value is directly comparable to the equivalent dose. However, it is worth noting that the signal intensity loss was equal to or lower (within 2%) than the dose loss. Such a small difference could be due to the fact that some aliquots were re-used multiple times over this experiment, which may have affected the grain's sensitivity."

>>We also added additional comments l. 294" Overall, sample SB44 bleaches faster than sample SB36. A difference in bleaching response from different K-rich feldspar samples has been observed by Sohbati et al., (2017) and interpreted as due to variation in the grain's optical transmission."
l.299: "Our results show the same tendency as the results reported by others (e.g., Bailif and Poolton, 1991; Spooner, 1993, 1994a, b, 2000; Sohbati et al., 2017). K-rich feldspar IRSL signal decay faster than the quartz OSL signal when exposed to yellow-orange light. The reason for such difference is, however, not fully understood. Additional analyses on well-characterized samples from different origins would be required to understand the relationship between bleaching rate and geochemical composition."